# A Comparison Between Single-Stage and Two-Stage 3D Tracking Algorithms for Greenhouse Robotics

**DOI:** 10.3390/s24227332

**Published:** 2024-11-17

**Authors:** David Rapado-Rincon, Akshay K. Burusa, Eldert J. van Henten, Gert Kootstra

**Affiliations:** Agricultural Biosystems Engineering, Wageningen University and Research, 6708 PB Wageningen, The Netherlands; akshaykumar.burusa@wur.nl (A.K.B.); eldert.vanhenten@wur.nl (E.J.v.H.); gert.kootstra@wur.nl (G.K.)

**Keywords:** deep learning, multi-object tracking, robotics, robotics in agriculture, transformers

## Abstract

With the current demand for automation in the agro-food industry, accurately detecting and localizing relevant objects in 3D is essential for successful robotic operations. However, this is a challenge due the presence of occlusions. Multi-view perception approaches allow robots to overcome occlusions, but a tracking component is needed to associate the objects detected by the robot over multiple viewpoints. Multi-object tracking (MOT) algorithms can be categorized between two-stage and single-stage methods. Two-stage methods tend to be simpler to adapt and implement to custom applications, while single-stage methods present a more complex end-to-end tracking method that can yield better results in occluded situations at the cost of more training data. The potential advantages of single-stage methods over two-stage methods depend on the complexity of the sequence of viewpoints that a robot needs to process. In this work, we compare a 3D two-stage MOT algorithm, 3D-SORT, against a 3D single-stage MOT algorithm, MOT-DETR, in three different types of sequences with varying levels of complexity. The sequences represent simpler and more complex motions that a robot arm can perform in a tomato greenhouse. Our experiments in a tomato greenhouse show that the single-stage algorithm consistently yields better tracking accuracy, especially in the more challenging sequences where objects are fully occluded or non-visible during several viewpoints.

## 1. Introduction

The agricultural and food sectors are increasingly under pressure from a rising global population and a concurrent decrease in available labor. Automation, particularly through the deployment of robotic technologies, has emerged as a solution in addressing these problems. However, the integration of robots into these sectors faces significant challenges, notably in the domain of robotic perception due to complex environmental conditions, such as occlusions and variation [1].

An accurate and efficient representation of the robot’s environment, including the relevant objects for a given task, is crucial for a successful robot operation in these environments [2,3]. Incorporating multiple views into a single representation has the potential to improve detection and localization even in highly occluded conditions [4]. For this, active perception (AP) algorithms are key in selecting optimal viewpoints [5]. However, building representations from multi-view perception requires associating upcoming detections with their corresponding object representations and previous measurements [3,6,7]. In object-centric representations, this task is often referred to as multi-object tracking (MOT).

A common robotic system that can be used for plant monitoring, maintenance and harvesting is a robotic arm with both a camera and a gripper placed in the end effector, as shown in Figure 1. This robot could be tasked with harvesting tomatoes in a greenhouse, which first requires detecting and localizing tomatoes in the plant. There are occlusions due to the tomatoes growing in trusses and the presence of leaves from the target plant and nearby plants. Therefore, to localize and properly estimate properties such as the ripeness of the tomatoes, the robot needs to collect multiple viewpoints along the plant and track the detected tomatoes over all the viewpoints. In this situation, the accuracy of the tomato tracking algorithm will determine the accuracy of the plant representation and, therefore, the ability of the robot to fulfill its task.

Recent advancements in MOT algorithms have led to the development of various approaches, prominently categorized into two-stage and single-stage methods. Two-stage methods, exemplified by SORT (Simple Online and Real-time Tracking) [8] and its more sophisticated successor, DeepSORT [9], separate the detection and re-identification (re-ID) feature extraction steps. In the first stage, objects are detected using a deep learning model. In the second stage, re-ID properties such as object position for SORT and/or re-ID black box features like in DeepSORT are generated for every object detected in the first stage. Single-stage methods, such as FairMOT [10], use a deep learning algorithm to simultaneously perform object detection and re-ID feature extraction within a single network inference step. This reduces the computational requirements and improves performance. However, this comes at the cost of increased algorithm complexity and a need for larger training datasets. In both two-stage and single-stage methods, the re-ID properties and features are used by a data association algorithm to match newly detected objects with those of previously tracked objects.

In agro-food environments, 2D two-stage MOT algorithms are the most commonly used approaches for tracking. They have been employed for tasks like crop monitoring and fruit counting [7,11,12,13,14,15,16]. The increased complexity and need for larger training datasets of single-stage methods, combined with the novelty of these algorithms, results in a lower presence of these methods in applied agro-food environments. In our previous work, we developed MOT-DETR (Multi-Object Tracking and Detection with Transformers) [17], a 3D single-stage MOT algorithm, and showed that it outperformed state-of-the-art MOT algorithms like FairMOT for tracking and monitoring tomatoes in greenhouses. MOT-DETR required approximately 50,000 training images, which is from 50 times [15] to 200 times [13] more data than the existing two-stage MOT algorithms. However, existing two-stage methods rely on object detection algorithms that were pre-trained on the COCO [18] dataset, which contains around 118,000 training images, while MOT-DETR does not.

In the tomato harvesting system described above and shown in Figure 1, the accuracy of MOT algorithms tends to decrease with the level of occlusion [7]. The amount of occlusion that the robot encounters in the sequence of viewpoints depends on two factors. The first being the amount of clutter in the environment, resulting in occluding leaves. The second factor relates to the motion of the robot. For example, if a robot arm moves at a low speed with the camera always pointing toward the same area of the environment, the resulting sequence will be similar to a video where the overlap between consecutive frames is large. However, due to obstacles in the environment and the presence of occluding objects, a robot arm might have to perform motions where the camera is seeing a completely different area of the environment for a while. This results in sequences where objects might disappear from the field-of-view (FoV) of the camera for a few viewpoints and where the difference between two consecutive viewpoint images is large. The resulting challenging sequences might require more powerful and complex MOT algorithms, like single-stage ones. However, this might come at the cost of increased complexity and amount of training data.

The performance differences and trade-offs of two-stage and single-stage MOT algorithms have not been widely studied in agro-food robotic environments. Hu et al. [14] compared several 2D single-stage and two-stage algorithms in a lettuce field using a wheeled robot. The sequence of images generated by the camera system of the robot contains a high overlap between consecutive viewpoints or frames due to the 2D motion of the wheeled robot. They show how, in this situation, two-stage algorithms perform better than single-stage ones. However, this conclusion might not be applicable to more complex scenes and 3D robot movements on arm-based robotic systems.

This paper presents a comparison of a 3D two-stage MOT algorithm, 3D-SORT [7], with a single-stage one, MOT-DETR [17], under different types of motions generated by a robot arm in a tomato greenhouse. The different motions represent different frame-to-frame distances and occlusions. Furthermore, we evaluated the performance of both algorithms when the viewpoints are generated using an active perception algorithm that aims to minimize occlusions across viewpoints [5].

## 2. Materials and Methods

### 2.1. Data

A dataset using five real plants from a tomato greenhouse was previously collected using the system shown in Figure 1-Left [17]. Per plant, viewpoints were collected using a planar motion sequence in front of each plant at a distance of 40 cm and 60 cm, as shown in Figure 1-Right. Per distance, a total of 600 viewpoints were collected following a planar path that covered an area of 60 cm height by 40 cm width in steps of 2 cm. In total, data from 5400 viewpoints from five plants were collected. Each viewpoint resulted in a color image of 960 by 540 pixels, a point cloud whose origin corresponds to the robot fixed coordinate frame, and the ground truth bounding boxes and tracking IDs of each tomato present in the viewpoint. Figure 2 shows example viewpoints from the same plant and truss at a distance of 40 and 60 cm between camera and plant.

The data were divided into train, validation and test splits as shown in Table 1. To prevent plants from being seen by the networks at train time and during the experiments, train and validation splits came from the same pool of four plants, while test splits were generated from a different plant.

### 2.2. Algorithms

In this work, we compared two 3D MOT algorithms, 3D-SORT [7] and MOT-DETR [17]. Both algorithms take as input the same data: a color image and its corresponding point cloud transformed into the robot’s fixed coordinate system. 3D-SORT, shown in Figure 3 (top), is a two-stage MOT algorithm, while MOT-DETR, shown in Figure 3 (bottom), is a single-stage MOT algorithm.

The 3D-SORT is a two-stage MOT algorithm that, at every viewpoint, detects objects and estimates their 3D location given a color image and a point cloud. 3D-SORT starts with a detection algorithm that detects objects in a color image. In this work, this step is performed with a YOLOv8 [19] network trained with the train set described in Table 1. Then, each tomato-bounding box is used to filter out the corresponding points from the point cloud. For each detected tomato, its corresponding points are used to estimate its 3D position with respect to the fixed robot coordinate frame. These 3D positions are used in the data association step to assign new detections with previously detected objects. The data association is performed using the Hungarian algorithm on a cost matrix calculated using the Mahalanobis distance between the location of newly detected tomatoes and the location of previously tracked tomatoes. Each tracked object, represented in blue in Figure 3 (top), contains a Kalman filter whose state represents the location of the object with respect to the robot. The associations generated by the Hungarian algorithm at every step are used to update the Kalman filter state of each tracked tomato using the 3D location of the associated detection. For more details of 3D-SORT, we refer to its original paper [7].

MOT-DETR is a single-stage MOT algorithm that uses the detection and tracking approach of FairMOT [10] but with the detection architecture of DETR (Detection with Transformers) [20]. Furthermore, it was extended to process simultaneously 2D images and 3D point clouds. MOT-DETR takes a color image and an organized point cloud, both with dimensions of 960 × 540 × 3. Features from the color image and point cloud are generated by two independent ResNet34 convolutional networks. These features are concatenated and passed to a transformer encoder-decoder following the DETR architecture. In the end, a set of multi-layer perceptrons is used to predict the object-bounding boxes and re-ID features using the output of the transformer decoder. For every viewpoint, MOT-DETR predicts the following outputs per detected object: 2D bounding box, class, and re-ID features. The data association is done by using the re-ID features of newly detected and previously detected objects to build a cost matrix using the cosine distance. This cost matrix is then passed to a Hungarian algorithm that generates the associations for tracking. We trained MOT-DETR as described in the original paper using the same data, which correspond to the train set depicted in Table 1 plus the synthetic dataset of the original paper. For more details, we refer to the original paper [17].

### 2.3. Experiments

We evaluated 3D-SORT and MOT-DETR under different sequence types that represent different motions and occlusion levels that a robot arm might encounter:**Sort.** For our test set, we used the available two sequences of 600 frames at two distances from the plant. For each sequence, 100 random viewpoints were randomly selected and sequentially ordered with the goal of minimizing frame-to-frame distance. The resulting sequence is similar to a video sequence with a low frame rate.**Random.** A total of 100 viewpoints were also selected out of the pool of viewpoints per sequence and plant. However, they were left unordered, which resulted in a sequence where jumps between frames were large and inconsistent, and objects were occluded during multiple frames.**AP.** Active perception improves the capacity of robots to deal with occlusions in unknown environments like tomato greenhouses [5]. Therefore, we studied the performance of our algorithm when 100 viewpoints were selected out of the pool of viewpoints using the AP algorithm developed by [5] that maximizes information gain between viewpoints. This method requires the definition of a region-of-interest (RoI). For this experiment, the region of interest was defined as a rectangular volume, approximately set around the middle-front of the robot where the stem of the tomato plant was expected.

Each sequence experiment was repeated five times by randomly selecting a different set of 100 random viewpoints, and the results were averaged. The MOT performance of the algorithms was evaluated using the High Order Tracking Accuracy (HOTA) [21], with its sub-metrics Localization Accuracy (LocA), Detection Accuracy (DetA) and Association Accuracy (AssA); and with Multi-Object Tracking Accuracy (MOTA) [22] and its sub-metric ID Switches (IDSW).

## 3. Results and Discussion

The results of our experiments can be seen in Table 2. 3D-SORT surpasses MOT-DETR in DetA and LocA. This outcome was anticipated, given 3D-SORT’s reliance on YOLOv8, a network pre-trained on large object detection datasets. In contrast to 3D-SORT, MOT-DETR’s simultaneous task on object detection and re-identification with a single network compromises its detection efficacy, as networks tasked with multiple objectives typically underperform those dedicated to a single task [10].

In terms of overall tracking accuracy, as measured by HOTA and MOTA scores, alongside AssA and IDSW, MOT-DETR consistently outperformed 3D-SORT across all tests. This performance difference underscores the ability of MOT-DETR to better understand a scene and the relationships between objects in it. The re-ID features generated by MOT-DETR are generated by a network that has access to the whole color image and point cloud data for each viewpoint. This means that relationships between the objects can be encoded by the network to boost tracking performance. This is in contrast to two-stage methods like 3D-SORT, where the re-ID properties and features are obtained from the object detection in 2D and/or 3D without considering any other information about the environment and nearby detections.

When the sequence of viewpoints was generated by an active perception algorithm, the tracking performance of both 3D-SORT and MOT-DETR increased, and while observing the sequences, the selected viewpoints had a lower level of occlusion in front of the plant stem, which is the region of interest of the AP algorithm. Furthermore, in the AP sequences, the tomato trusses are always in the center areas of the image, in contrast to the partially visible trusses in both Sort and Random sequences. This setup significantly reduces the likelihood of trusses and tomatoes being partially visible, thus mitigating detection and tracking challenges. This outcome shows the advantageous impact of AP in enhancing tracking accuracy under conditions where occlusions are prevalent.

## 4. Conclusions and Future Work

In this work, we have compared a two-stage 3D MOT algorithm, 3D-SORT, against a single-stage MOT algorithm, MOT-DETR. We showed how the two-stage method 3D-SORT yields better object detection results due to a more powerful object detection algorithm. However, the single-stage method, MOT-DETR, is able to consistently outperform 3D-SORT in overall tracking and data association performance. This shows that even with lower detection performance, the single-stage method is able to better understand the scene and encode the objects and their relationships. The difference in performance is larger with more complex sequences where frame-to-frame distance is large and objects might disappear from the camera FoV during several viewpoints. Furthermore, we showed how using active perception to reduce the number of occlusions present in the sequences boosts the tracking accuracy of both methods.

In future work, we plan to use and evaluate our tracking algorithms in a fruit harvesting and plant maintenance experiment. The tracking algorithm would be used to create complete and accurate representations of plants by associating information from multiple viewpoints. Then the plant representation would be used to define path trajectories and action points for the robot in order to perform harvesting and maintenance operations.

## Figures and Tables

**Figure 1 sensors-24-07332-f001:**
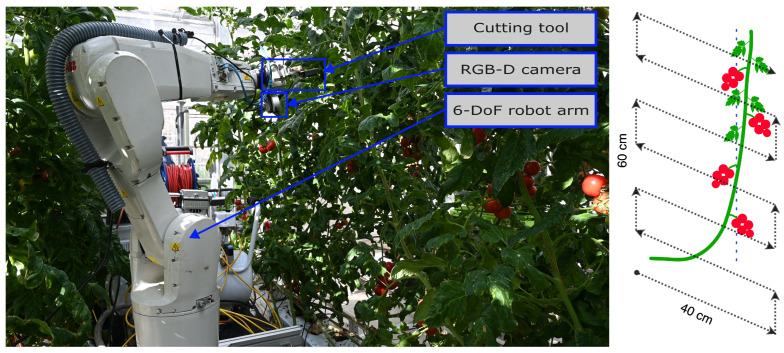
(**Left**). Robotic system used for data collection. We used a 6 DoF robot arm, ABB IRB1200. The robot is mounted over a mobile platform (not visible in the image) that allows movement over the greenhouse heating rails. On the end effector, we mounted a scissor-like cutting and gripping tool and a Realsense L515 camera. (**Right**). Illustration of the planar path followed by the robot with respect to the plant in front of it. An area of 60 cm (height) by 40 cm (width) was covered in steps of 2 cm.

**Figure 2 sensors-24-07332-f002:**
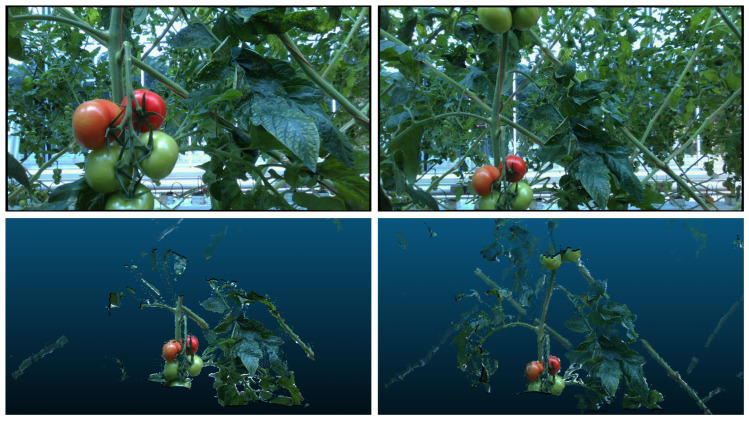
Examples of the images and point clouds collected by the robot. (**Left**). The distance from the camera to the plant is 40 cm. (**Right**). The distance from the camera to the plant is 60 cm.

**Figure 3 sensors-24-07332-f003:**
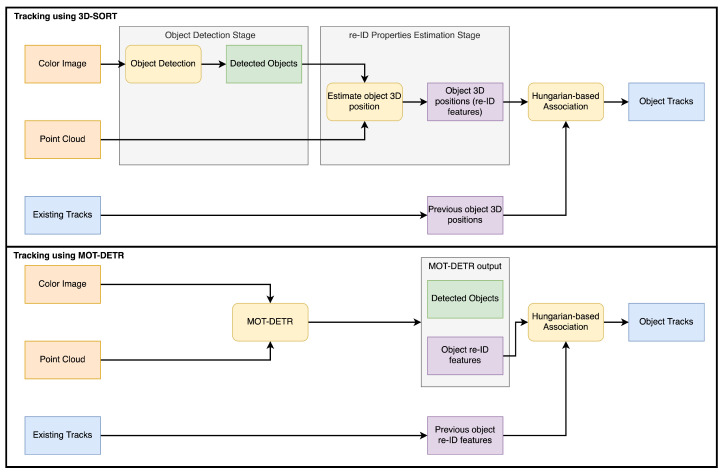
3D-SORT (**top**). First, the color image is processed by the object detection algorithm. The resulting detections are used together with the point cloud to generate a 3D position per detected object that corresponds to the re-ID property used by the data association step. The Hungarian algorithm is then used to associate the locations of newly detected objects with the previously tracked object positions. MOT-DETR (**bottom**). Color images and point clouds are used at the same time to detect objects with their corresponding class and re-ID features, which are black box features. The re-ID features are then passed to a Hungarian-based data association algorithm.

**Table 1 sensors-24-07332-t001:** Distribution of plants and images in train, validation and test splits.

Type	Split	# Plants	# Viewpoints
Real	Train/Validation	4	3570/630
	Test	1	1200

**Table 2 sensors-24-07332-t002:** Tracking performance results on different types of viewpoint sequences. Numbers in bold represent the best performance per metric and sequence.

Sequence	Algorithm	HOTA↑	DetA↑	AssA↑	LocA↑	MOTA↑	IDSW↓
Sort	3D-SORT	52.45	**65.59**	42.13	**92.79**	65.95	41.8
	MOT-DETR	**60.4**	56.3	**65.17**	79.17	**70.38**	**23.8**
Random	3D-SORT	33.37	**66.17**	16.92	**92.81**	46.24	200.2
	MOT-DETR	**59.63**	56.32	**63.44**	79.14	**66.86**	**57.4**
AP	3D-SORT	63.99	**79.89**	52.08	**93.21**	76.95	84.2
	MOT-DETR	**71.66**	66.08	**78.58**	80.16	**88.09**	**4.2**

## Data Availability

Data available on request.

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
