# Peer review of "A Comparison Between Single-Stage and Two-Stage 3D Tracking Algorithms for Greenhouse Robotics"

_sensors, 2024, doi:10.3390/s24227332_

Round 1

Reviewer 1 Report

Comments and Suggestions for Authors

This paper mainly compares two multi-object tracking (MOT) algorithms, 3D-SORT and MOT-DETR, reflecting the characteristics of single-stage and dual-stage 3D tracking algorithms.

The article is well-structured, logically clear, the methodology, and analysis of the results are clearly presented.

The recommendations are as follows: 

1.The article discusses "The increased complexity and need for larger training datasets of single-stage methods". It is recommended to give specific data volume requirements, especially at what scale of training sets does the MOT-DETR algorithm begin to surpass the 3D-SORT algorithm in overall tracking and data association performance.

2.Figure 1 depicts the flow of the two algorithms, and it is recommended to align the same parts in the upper and bottom parts to highlight the difference between the two methods. In addition, the stages marked as “MOT-DETR” in the MOT-DETR algorithm diagram can easily cause confusion.

Author Response

Comment 1. The article discusses "The increased complexity and need for larger training datasets of single-stage methods". It is recommended to give specific data volume requirements, especially at what scale of training sets does the MOT-DETR algorithm begin to surpass the 3D-SORT algorithm in overall tracking and data association performance.

Response 1. We thank the reviewer for their time and feedback. We agree that more details are needed as the comparisons are not straight forward. MOT-DETR was not pre-trained on COCO dataset, and uses 50000 training images. Existing two-stage methods rely on object detection models pre-trained on COCO (with 118000 training images) and normally require a few hundreds of images to fine-tune the object detection model. We added more details about this aspect on lines 63-67.

Comment 2. Figure 1 depicts the flow of the two algorithms, and it is recommended to align the same parts in the upper and bottom parts to highlight the difference between the two methods. In addition, the stages marked as “MOT-DETR” in the MOT-DETR algorithm diagram can easily cause confusion.

Response 2. We agree with the reviewer comment and we have modified the figure accordingly.

Reviewer 2 Report

Comments and Suggestions for Authors

The paper compares the performance of a 3D two-stage MOT algorithm, 3D-SORT, with a 3D single-stage MOT algorithm, MOT-DETR, in the context of 3D object tracking in a tomato greenhouse. These algorithms are evaluated on 3 different sequence types generated through Sort, Random and Active Perception algorithms on viewpoints. The paper investigates how each algorithm manages object occlusions and visibility changes across multiple viewpoints. The research results indicate that single-stage algorithm, MOT-DETR, provides better tracking accuracy, particularly in more complex sequences where objects are often fully occluded or disappear from view.

The paper clearly explains the aim of the research and the experiment results. The Related Work section appropriately discusses the object-tracking methods, and algorithms used in the previous research.

Including an additional figure in Figure 1 that illustrates the camera positions and other components mentioned (the scissor-like cutting tool and gripping tool on the robot arm) would enhance the description of the data collection procedure, and provide clear information about the hardware used.

The captions in Figures 1 and 2 are identical to those in the research paper [17], corresponding to Figures 4 and 5, respectively. Please paraphrase the caption of Figures 1 and 2 to avoid self-plagiarism.

The explanation of the data used in the paper could be enhanced by including figures that show a color image and the corresponding point cloud of the sample. Additionally, it would be helpful to specify the resolution of the color images and the size of the input data. Providing more detailed information in this section, supported by additional figures and diagrams, would strengthen the clarity of the section 2.1 Data.

Could you clarify the meaning of the 20 vertical steps and 30 horizontal steps shown in Figure 2? An explanation of this in the text would help improve understanding.

A brief overview of the methodology implemented in this paper would be beneficial. While 3D-SORT has been detailed in [7] and MOT-DETR in [17], a concise summary of these methods would provide helpful context for understanding their application in this study.

Could you indicate where the Kalman Filter is positioned in Figure 3?

Please review the first column of Table 2 to confirm if 'Dataset' is the correct label, or should it be changed to 'Sequence'?

Could you explain the future direction of the research in the Conclusion Section?

Please provide explanations for the acronyms used in the paper, such as "SORT," to improve clarity for readers.

A brief explanation of the metrics used to evaluate the results would enhance clarity. Additionally, if these performance metrics are associated with established algorithms, please include the relevant citations.

Author Response

Comment 1. Including an additional figure in Figure 1 that illustrates the camera positions and other components mentioned (the scissor-like cutting tool and gripping tool on the robot arm) would enhance the description of the data collection procedure, and provide clear information about the hardware used.

Response 1. We greatly appreciate the reviewer for the time spent and for the feedback. We agree with the comment that Figure 1 can benefit from text and arrows pointing to the relevant components. We added them. Furthermore, we also agree that the path followed by the robot is better shown in Figure 1. Therefore, we moved that illustration from figure 2 to figure 1.

Comment 2. The captions in Figures 1 and 2 are identical to those in the research paper [17], corresponding to Figures 4 and 5, respectively. Please paraphrase the caption of Figures 1 and 2 to avoid self-plagiarism.

Response 2. We thank the reviewer for the feedback. Based on this comment and the previous comment we have modified the figures and captions to fit better the specific work presented on this paper.

Comment 3. The explanation of the data used in the paper could be enhanced by including figures that show a color image and the corresponding point cloud of the sample. Additionally, it would be helpful to specify the resolution of the color images and the size of the input data. Providing more detailed information in this section, supported by additional figures and diagrams, would strengthen the clarity of the section 2.1 Data.

Response 3. We thank the reviewer for the feedback. In combination with previous feedback, we've modified Figure 2 to now show examples of color images and point clouds capture by the robot. We have added a better explanation of the figure at lines 108-109. Additionally, we have added the dimensions of the images at line 106. These modifications should improve the explanation of the data used for this work.

Comment 4. Could you clarify the meaning of the 20 vertical steps and 30 horizontal steps shown in Figure 2? An explanation of this in the text would help improve understanding.

Response 4. We thank the reviewer for pointing this out. We agree that a better explanation is needed. An explanation has been added on the caption of Figure 1 and on lines 103-104. We detected a typo in the previous version, as it's 30 vertical steps and 20 horizontal steps. This corresponds to an area of 60 cm (height) by 40 cm (width) covered on steps of 2 cm.

Comment 5. A brief overview of the methodology implemented in this paper would be beneficial. While 3D-SORT has been detailed in [7] and MOT-DETR in [17], a concise summary of these methods would provide helpful context for understanding their application in this study.

Response 5. Answered after next comment.

Comment 6. Could you indicate where the Kalman Filter is positioned in Figure 3?

Responses 5 and 6. We thank the reviewer for the feedback. We agree that the explanations of 3D-SORT and MOT-DETR are short. Therefore, we have extended the descriptions of both algorithms accordingly. We've also explained the Kalman filter for 3D-SORT and it's positioning on Figure 3. You can see all the changes on lines 130-134 and 138-143.

Comment 7. Please review the first column of Table 2 to confirm if 'Dataset' is the correct label, or should it be changed to 'Sequence'?

Response 7. We thank the reviewer for pointing this out. We've changed the label from "Dataset" to "Sequence".

Comment 8.  Could you explain the future direction of the research in the Conclusion Section?

Response 8. We thank the reviewer for the suggestion. We have included a future work area into the "Conclusions and Future work" section. It can be seen on lines 212-216.

Comment 9. Please provide explanations for the acronyms used in the paper, such as "SORT," to improve clarity for readers.

Response 9. We've included an explanation for the acronyms SORT, DETR and MOT-DETR.

Comment 10. A brief explanation of the metrics used to evaluate the results would enhance clarity. Additionally, if these performance metrics are associated with established algorithms, please include the relevant citations.

Response 10. We thank the reviewer for the feedback. HOTA and MOTA are standard MOT metrics. We've added the reference to the papers that introduced them where they're explained thoroughly and with their corresponding formulas.